# Post-Authorship Attribution Using Regularized Deep Neural Network

Abiodun Modupe [1,2,*], Turgay Celik [3], Vukosi Marivate [1] and Oludayo O. Olugbara [4]

1 Department of Computer Science, University of Pretoria, Lynnwood Road, Pretoria 0002, South Africa; vukosi.marivate@up.ac.za
2 School of Computer Science and Applied Mathematics, University of the Witwatersrand, Johannesburg 2000, South Africa
3 School of Electrical and Information Engineering, University of the Witwatersrand, Johannesburg 2000, South Africa; celikturgay@gamil.com
4 MICT SETA Center of Excellence in 4IR, Durban University of Technology, Durban 4001, South Africa; oludayoo@dut.ac.za
* Correspondence: abiodunmodupe@gmail.com

**Abstract:** Post-authorship attribution is a scientific process of using stylometric features to identify the genuine writer of an online text snippet such as an email, blog, forum post, or chat log. It has useful applications in manifold domains, for instance, in a verification process to proactively detect misogynistic, misandrist, xenophobic, and abusive posts on the internet or social networks. The process assumes that texts can be characterized by sequences of words that agglutinate the functional and content lyrics of a writer. However, defining an appropriate characterization of text to capture the unique writing style of an author is a complex endeavor in the discipline of computational linguistics. Moreover, posts are typically short texts with obfuscating vocabularies that might impact the accuracy of authorship attribution. The vocabularies include idioms, onomatopoeias, homophones, phonemes, synonyms, acronyms, anaphora, and polysemy. The method of the regularized deep neural network (RDNN) is introduced in this paper to circumvent the intrinsic challenges of post-authorship attribution. It is based on a convolutional neural network, bidirectional long short-term memory encoder, and distributed highway network. The neural network was used to extract lexical stylometric features that are fed into the bidirectional encoder to extract a syntactic feature-vector representation. The feature vector was then supplied as input to the distributed high networks for regularization to minimize the network-generalization error. The regularized feature vector was ultimately passed to the bidirectional decoder to learn the writing style of an author. The feature-classification layer consists of a fully connected network and a SoftMax function to make the prediction. The RDNN method was tested against thirteen state-of-the-art methods using four benchmark experimental datasets to validate its performance. Experimental results have demonstrated the effectiveness of the method when compared to the existing state-of-the-art methods on three datasets while producing comparable results on one dataset.

**Keywords:** authorship attribution; character embedding; bidirectional decoder; bidirectional encoder; deep learning; neural network; social media

## 1. Introduction

The ubiquity of portable computing devices, coupled with the globalization of the internet and social networking sites, have fundamentally transformed how humans generate and use the information for profitable ventures [1]. Myriad pieces of online text snippets and documents, such as blogs, tweets, and news are continuously generated from varying sources to deliver valuable and entertaining content to people anywhere [1]. The information is frequently posted on social networks every millisecond in the form of ideas, stories, opinions, sentiments, and other forms of expression [2]. Twitter, for instance, is a popular

microblogging service launched in the United States of America (USA) in 2006 [3] that has experienced significant growth in 2020, probably because of the coronavirus pandemic, with more than 330 million monthly active users at the beginning of 2019, as reported by Wikipedia [4]. Twitter is a famous platform with a huge volume of active users and a high revenue base, and its posts are widely called "tweets". A blog is a type of web page that a blogger usually maintains to transform user communication and overcome the unidirectionality of standard online communication [5]. There were approximately 600 million online blogs produced and read by about 77% of users as of July 2021 [6].

The present world is the era of "infobesity", where people have free access to post any kind of text snippets to convey their sentiments to the targeted audience [1]. However, posts are dismally misused to promote nocuous activities such as pirated software, terrorist communications, child pornography, and grossly belligerent, harassing, abusive, obscene, menacing, corrosive, and antediluvian messages [7–10]. Twitter was recently reported to be increasingly used for misinformation related to the pandemic and promised to start terming misleading tweets [11]. Text is a common material for posting news with a unique anonymity feature [12]. People hide their identities when posting sensitive information [13–15]. The delinquents usually masked their identities across multiple networks to circumvent the prowess of text-mining methods [16–18]. In addition, the use of proxy internet addresses, hashtags, mentioning of people in texts, and reposting of texts written by others is a common phenomenon [7,12,16]. The address of a genuine sender can be forged through an anonymous server to dispense offensive text messages across diverse pseudonymous channels [7,12,16]. The anonymity of social media posts, for example, imposes unique challenges and risks of tracing the identity of an offender without compromising privacy in cyberspaces [18–20].

The advancement of authorship attribution is supported by stylometric methods. The methods can be applied in a forensic process to identify the original author of a given anonymous post by analyzing the specific writing features of the writer [21–23]. Post-authorship attribution (PAA) is one method to turn the burden of a vast amount of online text snippets into a piece of useful knowledge. It is the study of unveiling the identity and sociolinguistic characteristics of an underlying author. The quantitative representation of the underlying features can be viewed as a fingerprint to quantify the writing style of an author [24]. The authorship attribution process has multifarious applications in the disciplines of computational linguistics [25], information retrieval [26], data mining [27,28], natural language processing [21], and personal electronic mail [29–31]. In addition, it is applicable for detecting plagiarism in an article. This is a scenario where, for instance, an editor of a journal wishes to identify whether a manuscript submitted for publication was written by someone else or has been published elsewhere [32–35]. Moreover, it is highly profitable in the marketing scenario for identifying the potential customers through their reviews [36,37], understanding the political preferences of audiences [38,39], and networks of political communication [40]. However, one of the principal challenges of authorship attribution is finding a suitable writing style that can adequately characterize a writer [41].

This study aimed to develop the method of regularized deep neural network for improving the accuracy of a PAA system. The unique contributions of this study to theory and practice are the following:

1.  The development of regularized deep neural-network method for improving the performance of a PAA system.
2.  The introduction of an interactive system to visualize the results of the PAA method, which could be useful in the process of an evidence-based forensic investigation.
3.  The demonstration of the performance of the proposed PAA method through experimental comparison with existing prominent methods.

The content of this paper is succinctly organized as follows. Section 1 provides the introductory message. Section 2 discusses the related studies in chronological order. Section 3 designates the proposed method. Experimental results, including datasets, parameters

settings, discussion of results, and visualized results are explicated in Section 4. We provide discussion in Section 5 and concluding remarks are provided in Section 6.

## 2. Related Works

Research on the authorship-attribution process has generally been exhilarating for a long time. Manifold studies have presented different methods of developing automated systems for identifying the genuine author of a text. In most cases, authorship attribution can be effectively accomplished in two stages. The first stage is related to the extraction of stylometric features from the original post to characterize the unique writing style of an author [21]. Stylometric features can be compactly organized into three main categories of lexical features, syntactic features, and structural features [7,21–24]. Lexical features are statistical measures of text characters and word-based lexical text variations such as vocabulary richness and word length distribution [21]. Syntactic features capture the universal part of speech (POS) tags from the text structure [42]. Structural features are associated with the organization of texts, such as the average number of words in a sentence, and the use of indentations or paragraphs [43]. The second stage of the authorship-attribution process is allied with the training of machine-learning methods to learn the extracted stylistic features for finding the boundaries between the predicted author, and an anonymous writer through the minimization of a loss function [23].

The connotation of feature extraction is paramount to the task of authorship attribution. Past studies have found that character-level n-grams are among the most compelling features for effectively identifying authors [32,44–47]. The authors in [44,45] used the common n-gram model to capture a representation of the writing style of a writer who translated text between the English language and Greek language. The byte-level n-gram profile was applied in [46] to identify the author of a source code. The writing style of an author was captured in [47] using a set of local histograms (LH) over character n-grams. The researchers achieved an accuracy of 86.4% from Reuters corpus volume 1 (RCV1) [48] comprising English journalism documents authored by 10 and 50 writers, refer to as CCAT10 and CCAT50, respectively.

The researchers in [49] used 10,000 blogs for the task of authorship detection, in which 500-word snippets, one for each author, were considered as test examples. The 20% to 34% of the texts were classified with an average accuracy of 80% using the linear support-vector machine (SVM) while the remaining texts were considered unknown. In a separate study using the same dataset, the authors in [50] used character n-grams to effectively capture the stylistic and morphological information of the authors. They employed the similarity-based method to attribute 500-word snippets to 1000 authors to achieve a high precision rate of 93.2%. The survey of a variety of feature types and categorization methods proposed in the past for authorship attribution based on literature corpus, email and blog corpora was reported in [22]. The SVM and Bayesian regression methods were experimented to achieve 80% on the 1000 most frequent words and to 86% on 1000-character n-grams with the highest information gain.

The advantage of using character-level n-gram features was attributed to the high priority of sub-word features for authorship attribution [51]. The authors in [52] used tensors of second order to represent the stylistic features of texts with 2500 most frequent 3-g. The importance of punctuation marks for cross-domain authorship attribution of the PAN 2019 competition that comprised the identification of the authors of fan fiction was explicated in [53]. The application of an n-gram model with everything except for punctuation marks masked by an asterisk symbol was shown in [54]. The authors in [55] claimed that digits and named entities are important identifiers of the writing styles of authors. The application of disjoint author–document topic (DADT) [56] over the author topic (AT) [57] and latent Dirichlet allocation (LDA) [58] was used to extract domain-indicative lexical items from a text. The work achieved a similar purpose because text representations generated from n-gram features tend to be of higher dimension, and sparse [59]. However, measuring the choice of the vocabulary of an author is insufficient

to adequately capture the extreme intertextual and intertextual variation in style-related tasks [60].

There are different subtasks of authorship attribution that heavily depend on the nature of an input text. They include text-authorship attribution (TAA), which generally involves a text document comprising several words [21,22]. This subtask is commonly found in digital publications and data-mining applications [23]. The subtask of code-authorship attribution (CAA) is to identify the real author of a given computer-programming code. The application of CAA includes software forensics [61], detection of malicious code [61,62], detection of code plagiarism [63–65], and learning software ownership [66]. The subtask of post-authorship attribution (PAA) generally involves short snippets with about 500 words that are commonly found on social networks and the internet [26].

Most of the past studies on authorship attribution have generally focused on long texts. However, because of the proliferation of web applications, widespread acceptance of social media networks, and advancement in internet technology, an increasing volume of studies are focused on developing models for improving the interpretability of online messages [18,67]. The authors in [68] used the methods of common n-grams (CNG) [69], source code author profile (SCAP) [44–46], and recentered local profiles (RLP) [70] to weight the profiles of authors on a collection of character-level $n$-gram features. They achieved an accuracy of 54.93% with the character $n$-gram length, $n$ using the RLP on an Internet relay chat (IRC) dataset. The character and word n-grams were used in [71] as a K-signature to interpret the unique writing style that appears in the least K% tweets posted by an author. They achieved an accuracy of 50.7% on a dataset containing 50 authors with 50 training tweets per author. Moreover, an accuracy of 71.2% was achieved with 10,000 training tweets per author. The accuracy of 20.25% was reported in [72] using the naive Bayes (NB) method on 2000 messages taken from an SMS corpus with 81 authors and a maximum of 50 messages per author.

The authors in [73] achieved an accuracy of 53.2% in identifying an author from 10,000 scale microblog users using character n-gram frequency with cosine similarity to discover the most relevant stylometric features. The authors in [74] used lexical, syntactic, tweet-specific, and other useful features to extract stylometric information from a given tweet using a natural language toolkit (NLTK). The features corresponding to an anonymous author were identified with the help of the SVM method. The researchers achieved an accuracy of 64.54% with 300 most recent tweets of 20 authors. The researchers in [19] presented an exhaustive bibliography survey of the state-of-the-art (SOTA) techniques for PAA tasks. They employed shallow classifiers such as support vector machine (SVM) and random forest (RF) allied with a bag-of-words (BOW) feature representation. They explored the performance dependency of their proposed method with multiple authors and messages. However, their results are still far from satisfactory, despite advancing the SOTA for PAA tasks. The authors presented a classification accuracy of less than 65% in a new experimental dataset of 50 authors. This number plummets to less than 45% in the context of 1000 authors, showing that PAA is still an open research endeavor that requires further improvement.

The authorship attribution has lately been built on deep neural networks by researchers [60,75–77]. The sequence of words or a string of characters is mainly used as input to the neural networks. Most methods focus on lexical features, even though lexical-based language models are not scalable when dealing with authorship containing diverse topics [78,79]. The syntactic characteristics are content-dependent, more robust, and effective against the topic variance. The authors in [60] used continuous representation through a neural network combined with a classification layer for authorship attribution. They reached an accuracy of 94.8% on the IMDB62 dataset [80]. The authors in [75] applied a CNN model for a large-scale task of authorship attribution based on the work reported in [81]. They evaluated their method on different text data such as emails, reviews, blogs, and tweets to obtain excellent accuracy results between 85.0% and 95.0% across the datasets used. Three language methods of lexical, syntactic, and character-level features were com-

bined in [1] for authorship identification using the international conference on weblogs and social media (ICWSM) labeled Twitter datasets [82] to achieve the best accuracy of 96.3%. The authors in [76] used CNN based on a sequence of character n-grams for the authorship-attribution problem of short posts. They evaluated their method using the dataset presented in [71]. Besides showing a satisfactory result of 76.1% of accuracy for 50 authors, they noticed that nearly 30.0% of the users presented in the dataset behaved as automated bots, turning the task into a straightforward method. However, when the profiles were removed, the classification accuracy dropped to 68.3% for 35 authors.

The researchers in [83] tackled the problem of authorship attribution using a single author from an anthology reference corpus [84] and POS tags with the application of a CNN method. They reached a better generalization by replacing uncommon words with their POS tags to achieve good results from a scientific publications corpus. The authors in [85] used different types of embedding structures based on character, word, *n*-gram, and POS tags in a CNN model to learn the representations of writing styles of authors of Twitter and Weibo posts. They adopted the dataset in [71] to obtain accurate results for 50 authors below the results reported in [76]. The authors in [77] introduced a syntax encoding using the CNN collaborated with lexical features for authorship attribution. They utilized a syntax parse tree of sentences to achieve accuracy results of 81.00%, 96.16%, and 56.73% on the CCAT50, IMDB62, and Blogs50 datasets, respectively. The authors in [78] introduced syntactic and lexical representations using the attention-based hierarchical neural network to encode the syntactic, and semantic structures of sentences in documents to respectively achieve the accuracy results of 82.35% and 73.83% on CCAT50 and Blogs50 datasets [86].

The authors in [87] used a method based on pretrained language models such as bidirectional encoder representations from Transformers (BERT) [88], pretraining a general-purpose language representation of DistilBERT [89], and a robustly optimized BERT pre-training (RoBERTa) [90] on a large collection of highly diverse authorship-attribution datasets to provide better generalization results. Previous works use this family of models combined with a dense layer and a SoftMax function to extract stylometric features of character-level bi-gram and trigram in an ensemble model [39]. They achieved an accuracy of 93.0% at 10 epochs that did not perform well on IMDB62 datasets [80]. Furthermore, the method reached a relative improvement of 5.3% accuracy on a Blog dataset with 50 authors compared to the previous techniques [77]. Table 1 provides a comparison of related studies based on common characteristics.

**Table 1.** Comparison of related studies based on common characteristics.

| Reference | Features | Representation | Classification Method | Visualization | Data |
|---|---|---|---|---|---|
| [1] | Lexical, syntactic, and topical | Word2Vec, and Doc2Vec | Shallow NN and SoftMax | No | Twitter |
| [19] | Lexical, syntactic, and structural | BOW over n-grams | SVM, RF, and SCAP | No | Twitter |
| [22] | Stylometric | Character n-grams and word | SVM and Bayesian | No | Literature corpus, email, and blog posts |
| [32] | Lexical | Byte-level n-grams | Dissimilarity measure | No | English and Chinese |
| [35] | Lexical and syntactic | BERT | Dense, SoftMax | No | Email, IMDB, and blog posts |
| [78] | Lexical, syntactic, and structural | CNN, LSTM, and attention | SoftMax | No | CCAT10, CCAT50, Blogs10, and Blogs50 |
| [45] | Lexical | Byte-level n-grams | Dissimilarity Measure | No | Source code |

**Table 1.** *Cont.*

| Reference | Features | Representation | Classification Method | Visualization | Data |
|-----------|----------|----------------|-----------------------|---------------|------|
| [46] | Lexical | Character-level n-grams | SVM | No | CCAT10 |
| [49] | Lexical and syntactic | TF-IDF over word, character | SVM | No | English blog posts |
| [50] | Lexical and syntactic | Character-level n-grams | Similarity-based methods | No | Blog posts |
| [51] | Lexical and syntactic | Character-level n-grams, affix punctuation 3-g | SVM | No | CCAT10, and CCAT50 |
| [52] | Lexical and syntactic | Frequent 3-g | SVM | No | CCAT10, and CCAT50 |
| [53] | Lexical and syntactic | TF-IDF over n-grams | Soft voting ensemble | No | PAN 2019 |
| [55] | Lexical and syntactic | Punctuation marks, name entity, and character n-grams | SVM | No | CCAT10 |
| [61–63] | Lexical syntactic, and structural | LDA, AT, and DADT | Probabilistic Model | No | Judgment, email, and IMDB |
| [60] | Lexical, syntactic, and structural | Character, word, and FastText | SoftMax | No | Judgment, IMDB62, and CCAT50 |
| [68] | Lexical | Weighting n-gram | CNG, SCAP, and RLP | No | IRC, and Twitter |
| [71] | Lexical and syntactic | Character, Word n-gram, and K-signature | Linear SVM | No | Twitter |
| [72] | Lexical | ChatSafe | NB | No | SMS |
| [73] | Lexical and syntactic | POS tag combine n-grams | Cosine Similarity | No | Twitter |
| [74] | Lexical, syntactic, and structural | NLTP | SVM | No | Twitter |
| [75] | Lexical and syntactic | Character and word-level n-grams | Word embedding, CNN, and SoftMax | No | Email, Blogs, Redditt, and Twitter |
| [77] | Lexical and syntactic | Syntax Parse Tree | CNN, SoftMax | No | CCAT10, CCAT50, IMDB, and Blogs |
| [83] | Lexical and syntactic | POS tag and word embedding | CNN, SoftMax | No | ARC |
| [85] | Lexical, syntactic, and structural | Character, word, n-gram, and POS tags | CNN, SoftMax | No | Twitter and Weibo |
| [87] | Lexical, syntactic, and structural | BERT, RoBERT, and DistilBERT | SoftMax | No | CCAT10, IMDB, and Reddit |
| **RDNN** | **Lexical, syntactic, and structural** | **Character-level CNN and DHN-BLSTM** | **Dense and SoftMax** | **Yes** | **CCAT50, IMDB62, Blogs50, and Twitter50** |

The present study focused on the PAA subtask, which is generally hard to solve because of the inherent characteristics of the input text. The input texts are generally casual snippets with short contents and rife with obfuscating vocabularies such as idioms, onomatopoeias, homophones, phonemes, synonyms, acronyms, anaphora, and polysemy. However, it can be noted that none of the few previous studies on PAA had focused on interpretability, which is a critical factor for real-life investigation scenarios. This

current study further addresses the knowledge gap by adopting a method that utilized the extracted features to visualize the results of the authorship attribution. Multiple studies have strikingly reported a higher authorship-attribution accuracy with several style markers such as lexical, syntactic, idiosyncratic, and structural markers [19,23]. However, the efficiencies of most of the past methods decreased with an increasing number of authors [26]. They significantly depend on extensive feature engineering to reflect the content and style of an author [1,21,30]. They lacked transparency and interpretation capability [91] which are essential for language-based pretrial investigation.

The regularized deep neural network (RDNN) is introduced in this paper for the task of PAA to achieve a better classification performance with a much faster convergence speed. The RDNN method promises to construct stylometry features by employing CNN, bidirectional long-short-memory (BLSTM), and distribution highway network (DHN). The purpose was to realize an improved PAA system with a high-precision visualization capability when compared to the SOTA methods. The DHN guides the sentence structure in the LSTM cell to speed up the convergence rate of the BLSTM layers. The novel method, unlike the existing methods, can learn abnormal terms from combinations of characters, such as misspellings, emoticons, and stacks on two BLSTM layers. In addition, the BLSTM decoder translates the exact text in reverse order to produce output, and the BLSTM encoder is used to affirm the state of the past context of the text from a character-level CNN. Moreover, the DHN will allow the seamless flow of unimpeded information across the network before reaching the classification layer. The RDNN method was tested on four different datasets of text snippets to investigate its generalization capability across varying input modalities.

## 3. Proposed Method

The principal objective of this study was to develop the method of regularized deep neural network (RDNN) for improving the accuracy of a PAA system. The method has been implemented in a three-layer architectural PAA system (Figure 1). The system consists of a CNN character-level layer, distributed highway network (DHN) with bidirectional long short-term memory (BLSTM) DHN-BLSTM layer, and a feature-classification layer as explicated in this section. The purpose of agglutinating the CNN character-level, DHN, and BLSTM layers was to achieve a PAA system with a better classification performance, a much faster convergence speed, and high-precision visualization capability.

### 3.1. CNN Character-Level Layer

The CNN character-level layer landscapes three essential components, which are character embedding, temporal convolution, and max-over-time pooling. Character embedding is a numeric-vector representation for a word constructed from the character n-grams. The PAA system receives as input a sequence of characters of a post. It finds the corresponding one-hot vector representation for each character through a dictionary of $m$ characters. The maximum length of a post is 512 characters, while the maximum number of sentences in a document is bounded at 280 because of variations in the lengths of posts. If the maximum length of a post surpasses 512 characters, the sequence is padded at the preprocessing stage to have the same length because truncating the sequence will yield worse performance [75]. English-language training datasets were utilized in this study. The dictionary that supports embedding contains 70 English characters, which include 26 letters, 10 numeric letters, and 34 other letters such as white space as follows:

abcdefghijklmnopqrstuvwxyz-,;.!? : "'/\|_@#$%&* ~ '+ =<> ()[]{}0123456789

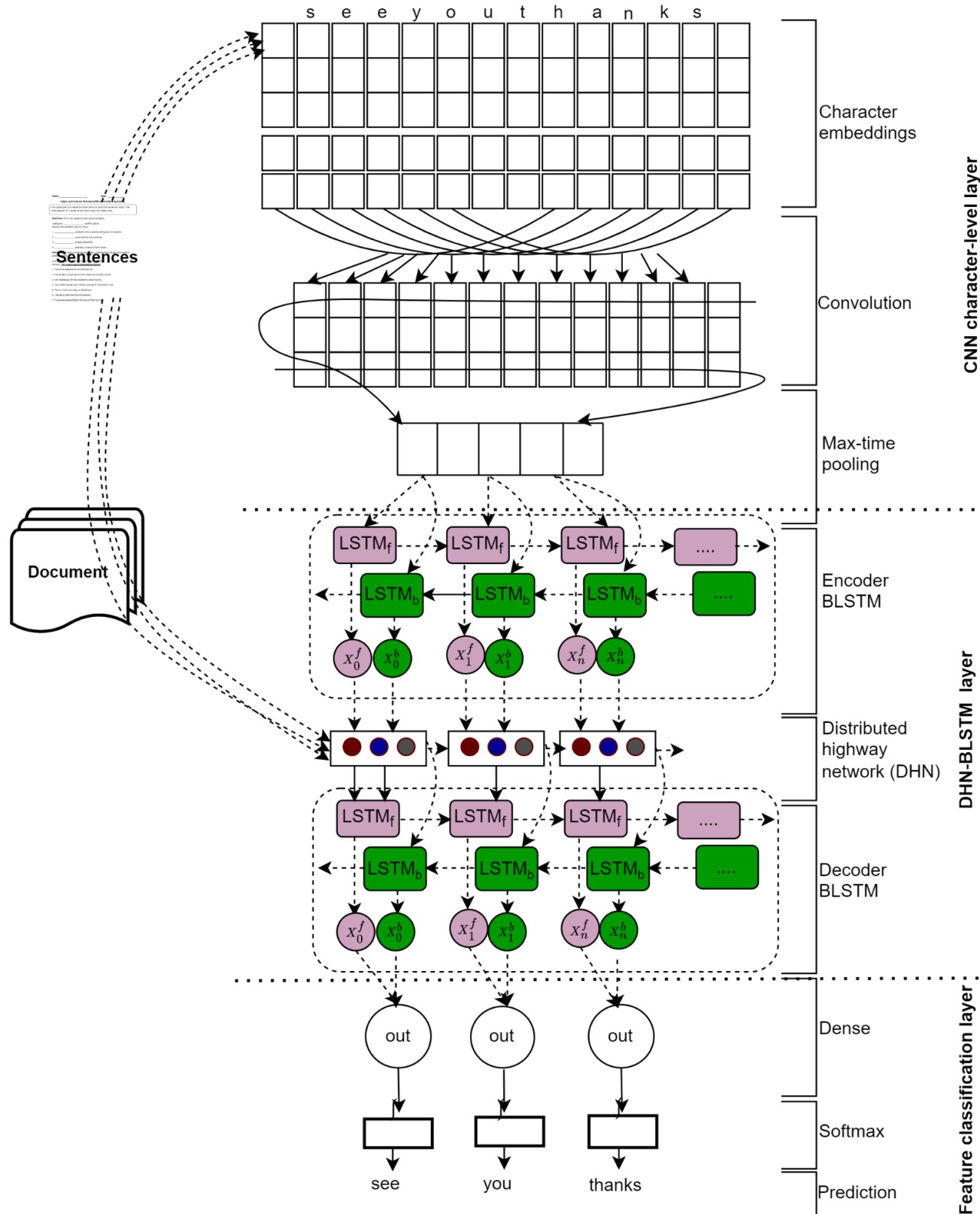

**Figure 1.** The architecture of the RDNN PAA system.

The CNN character-level layer of the proposed RDNN method is a slight variant of the deep CNN character-level [92]. In the CNN layer, each word in a post is replaced with its one-hot vector representation. Hence, we denoted $V$ to be the fixed vocabulary size of $d$ being the dimensionality of character embedding, and $X \in \mathbb{R}^{d \times |V|}$ to be the matrix

of character embeddings. Suppose a word, $w \in V$, of length $l$ is made up of a sequence of characters $[c_1, c_2, \ldots, c_l]$, the character-level representation of $w$ is given by the matrix $C^w \in \mathbb{R}^{d \times l}$, where the $j$-the column corresponds to the character embedding $c_j$ is the $c_j$-th column of matrix $X$. The $C^w$ matrix can be padded with zero for batch processing to ensure that the number of columns is equal to the maximum length of a sentence for all words $V$.

The temporal convolution exhibits longer memory than recurrent networks and is applied in this study to give better performance for character-sequence modeling. The convolution operations are performed between the input matrix and multiple filter kernels with stacking to form the input matrix. A new narrow convolution is used between $X^w$ and a kernel $H \in \mathbb{R}^{d \times \omega}$ as a filter. Here $\omega$ is the width after adding a bias $b$ and applying a nonlinearity to obtain $f^w \in \mathbb{R}^{l-(\omega-1)}$ a feature map. The $i$th element of $f^w$ is explicitly defined as follows.

$$f^w[i] = \tanh(X^w[*, i : i + \omega - 1], H + b) \tag{1}$$

Here, $X^k[*, i : i + \omega - 1]$ is the $i$-to-$(i + \omega - 1)$-th column of $X^w$ and $P, S = Tr(PS^T)$ is the Frobenius inner product operation.

The max-over-time pooling operation is ultimately applied to obtain a fixed dimensional representation of a word, which is the output of the DHN-BLSTM layer. The max over time of Equation (2) was taken as the feature corresponding to the filter $H$ to extract the highest value for a given filter. Each filter is essentially picking out a character n-gram with a size of the n-gram that corresponds to the filter width. Finally, we take the max-over-time as follows.

$$y^w = \max_i f^w[i] \tag{2}$$

### 3.2. Distributed Highway Network with Bidirectional Long Short-Term Memory

The distributed highway network (DHN) and bidirectional long short-term memory (BLSTM) are essential components in the second layer of the RDNN PAA system. The DHN features between the encoder BLSTM and decoder BLSTM are to guarantee the validity of the network information. Moreover, they guide the sentence structure in the LSTM cell to speed up the convergence rate of the BLSTM layer. Different from the existing models, DHN is added to the BLSTM to effectively regulate the flow of information and reduce the number of model parameters. The encoder BLSTM was employed to find the state of the past context of a post from a character-level CNN. The decoder BLSTM will translate the exact post in reverse order and produce output.

#### 3.2.1. Distributed Highway Network

A distributed highway network (DHN) can optimize a deep-learning model with the core purpose of regulating the paths for information to follow across the different layers with a gate mechanism in LSTM [93]. In a feedforward neural network (FFNN) consisting of $L$ layers, each layer can use the nonlinear transformation $F$ with a weight matrix $W_F$ to generate the output $y_i$ for the input $x_i$. The tensor $y$ can be represented as follows:

$$y = F(x, W_F) \tag{3}$$

The DHN introduces two nonlinear transforms, $N$ and $T$, in Equation (4), so that the output $y$ can be rewritten as follows:

$$y = F(x, W_F) \cdot N(x, W_N) + x \cdot T(x, W_T) \tag{4}$$

$F$ is an affine transform followed by a nonlinear activation function, $N$ is the transform gate, and $T$ is the carry gate that expresses how much output can be produced by transforming and carrying the input. $T$ is usually set to $N - 1$ in Equation (4) to obtain Equation (5) as follows:

$$y = F(x, W_F) \cdot N(x, W_N) + x \cdot (1 - N(x, W_N)) \tag{5}$$

The dimensionality of $x, y, F(x, W_F)$ and $N(x, W_N)$ must be the same to guarantee the validity of Equation (5). Consequently, based on the output of the transform gates, a highway layer can smoothly vary its behavior, and the outcome is used to adjust the vector from the CNN character-level layer.

### 3.2.2. Bidirectional Long Short-Term Memory

Long short-term memory (LSTM) captures the contextual information for solitary text sequences. However, the feature representation that is obtained from the CNN character level does not contain the sequence information. The bidirectional LSTM (BLSTM) is specialized in sequential modeling and can further extract the contextual information from the feature sequences obtained in the convolutional layer. Hence, we have used it to learn the representation of long-range dependencies based on the surrounding word distributions. The encoder BLSTM was used to capture the lexical, and syntactic information from the CNN character level. The one-dimensional convolution involves a filter sliding over a sequence to account for features at different positions. Thereafter, the decoding layer operates on the encoded sentence representation with DHN. The decoder BLSTM will find the hidden structural representations for authorship attribution. The forward LSTM network computes the state $\overrightarrow{h_t}$ of the past left context of the sentence $c_j$, while the backward LSTM network reads the same sentence in the reverse order to produce $\overleftarrow{h_t}$ given the future right context. The $\overrightarrow{h_t}$ and $\overleftarrow{h_t}$ outputs are concatenated to obtain the final output of the BLSTM network as follows.

$$h_t = \left[ \overrightarrow{h_t} : \overleftarrow{h_t} \right] \tag{6}$$

The number of hidden layers for the input text is set to $n$, and the result of the decoder BLSTM network can be expressed as follows.

$$\mathbb{H} = [h_t, h_2, \ldots, h_m] \tag{7}$$

The parameter $m$ is the length of the input post. The output of the last LSTM layer is $\mathbb{H} \in \mathbb{R}^{m \times (2 \times n)}$ with each row of the matrix $\mathbb{H}$ representing a feature vector of the sentence generated by BLSTM. The hidden states of the time steps of BLSTM or the last layer of the time step hidden states of multilayer BLSTM are concatenated to obtain the feature representation of a post. The output of the highway network is used as input to the decoder BLSTM to learn the semantic representation of post sentences. The hidden document embedding of BLSTM is fed into a fully connected network for training and classification. The additional feature exploration was carried out to analyze the most important features for the overall performance of the RDNN method. The Shapley additive explanation (SHAP) [94] and local interpretable model-agnostic explanation (LIME) [95] have been applied in this study for interpretability.

### 3.3. Feature-Classification Layer

The feature-classification layer consists of a dense layer or deeply connected layer from its preceding layer, SoftMax function, and prediction module. The feature-vector representation of a post is fed into a fully connected output layer that maps the weight vector of the decoder BLSTM to obtain the correct post classification as follows:

$$y(C_t | *) = \text{softmax}(\mathbb{H}_t, h_{t-1}) \tag{8}$$

Here, $C_t$ is the feature vector for a post, the weight matrix $h_{t-1}$ is a dictionary containing the sentence embeddings learned for each feature, and $\mathbb{H}$ is a weight matrix that is learned to predict the appropriate class label.

The entire network is trained as an end-to-end model, such that a post written by the same author would result in small values as shown in Equation (8). Hidden feature vectors in Equation (7) must be made insensitive to topical variations between posts. The

model is practically trained on randomly selected English language datasets without external resources. The external resources include WordNet or pretrained embeddings to control the generalization of error from the model [96,97]. A dropout mechanism is applied to the regularization after the penultimate layer to learn the model parameters. The following cross-entropy loss is minimized during the training using the AdamW optimization algorithm [98].

$$\text{CrossEnt}(p, q) = -\Sigma p(d) \log(q(d)) \tag{9}$$

Here, $p$ is the true distribution of one-hot vector representation of post characters and $q$ is the output of the SoftMax. This inherently corresponds to computing the negative log probability of the true authors.

## 4. Experimental Results

The performance of the RDNN method is experimentally evaluated on large-scale English text datasets. The experiments were implemented using Ubuntu 18.04, Python 3.7 with Keras TensorFlow 1.15.1 application programming interface (API) on an Intel i7 4.0-GHz CPU computer with 64G DDR4 memory.

### 4.1. Datasets

The experimental datasets of this study are CCAT50, IMDB62, Blogs50, and a new Twitter50 dataset congregated using the Twitter application programming interface (API). The first three datasets have been extensively used in numerous previous studies [48,80,86]. CCAT50 is a collection of international texts and television news in the English language with different topics on business, politics, and industrial documents written by 50 authors [48]. It is a larger version of the CCAT10 with 5000 documents from each of the authors. IMDB is a prolific user dataset composed in May 2009 from the internet movie database at www.imdb.com [80]. The dataset includes 184 authors, each with at least 500 movie reviews per author. The authors write largely movie-related board posts, but some are about television, music, and other interesting topics. In this study, we used a subset of the IMDB dataset of prolific writers, called IMDB62 [99], which contains 62,000 reviewers of 1000 reviews per user. The reviews of each author were obtained using proportional sampling without a replacement. That is, for each author, 1000 reviews have the same rating frequencies as their complete set of reviews. The Blogs50 dataset is made of posts written in August 2004 by 50 top bloggers from the dataset of 681,288 blogs written by 19,320 bloggers from blogger.com (accessed on 16 February 2022) [86]. Twitter50 is a new dataset of tweets that was divided into training and testing sets to benchmark our PAA method with the SOTA methods. We created the dataset using the Twitter API to aggregate a list of 1391 celebrities and influencers from 68 different knowledge domains, including politics, technology, and arts. The summary description of the experimental datasets are concretely provided in Table 2.

**Table 2.** Description of the experimental datasets.

| Datasets | Classes | Word Size | Character Size | Average per Author | Total Size |
|----------|---------|-----------|----------------|--------------------|-----------| 
| CCAT50 | 50 | 584 | 3010 | 100 | 5000 |
| IMDB62 | 62 | 345 | 1742 | 542 | 62,985 |
| Blogs50 | 50 | 117 | 542 | 682 | 681,288 |
| Twitter50 | 50 | 36 | 119 | 270 | 1,109,964 |

### 4.2. Parameter Settings

The kernel sizes were 3, 4, 5, and 7 with the corresponding channel numbers of 64, 128, 256, and 512, respectively. The ReLU activation function was used for a fair comparison. The dropout rate varied from 0.1 to 0.7, and the mini-batch size was set to 32.

The number of hidden layers, m, in the DHN0-BLSTM layer was set to 512. The length of each convolutional layer was set to 128 for each kernel size. The SoftMax activation function was used in the feature-classification layer, and character embedding was initialized using the Glorot uniform initialization [100]. The entire model was trained for 60 epochs using the AdamW optimizer [98] instead of Adam to yield a better training loss because it substantially generalizes better than Adam [101]. The learning rate was set to $1 \times 10^{-4}$, and the dropout rate varied from 0.3 to 0.7. The number of layers in the DHN was set to 2. Table 3 reports all the hyperparameters in CNN-DHN-BLSTM architecture on all the different datasets. Here, d is the dimensionality of character embeddings, $f, \sigma$ are nonlinearity functions, $p_s$ is the pooling size, $\delta_o$ is the dropout, $l$ is the number of layers, and $m$ is the number of hidden units.

**Table 3.** The setting of hyperparameters in RDNN method on different datasets.

| Layers | Parameters | CCAT50 | IMDB62 | Blogs50 | Twitter50 |
|---|---|---|---|---|---|
| CNN Character-level | $d$ | 30 | 70 | 70 | 70 |
| | $f$ | ReLU | ReLU | ReLU | RELU |
| | $\delta_o$ | 0.5 | 0.5 | 0.5 | 0.5 |
| | $p_s$ | 2 | 2 | 2 | 2 |
| BLSTM | $m$ | 128 | 128 | 256 | 512 |
| FCL | $l$ | 2 | 2 | 2 | 2 |
| | $\delta_o$ | 0.7 | 0.5 | 0.5 | 0.5 |
| | $m$ | 512 | 512 | 128 | 128 |
| | $\sigma$ | SoftMax | SoftMax | SoftMax | SoftMax |

*4.3. Results*

The RDNN method is here compared to numerous SOTA methods for PAA with results presented quantitively and visually. The 5-fold cross-validation method with Keras tuner was used to find the best hyperparameters for the RDNN method. It utilizes 10% randomly split off as a stratified test set to minimize computational expensiveness that can occur with a higher number of folds. The best hyperparameters used to optimize the RDNN method are summarized in Table 3. A seed was used to reduce randomness by a further 10% for each training set. The quantitative performance metrics were the accuracy, F1-score, and loss curve calculated over 60 epochs with a 0.8/0.1/0.1 train/validation/test split. The visualization analysis was performed using the SHAP [94] and LIME) [95] for interpretability.

4.3.1. Quantitative Results

The quantitative results based on accuracy and F1-score performance measures are shown in Table 4. Most of the methods did not compute the measures across the four experimental benchmarked datasets. The RDNN method recorded good accuracies and F1-scores for three of the experimental datasets when compared to other SOTA methods. The proposed method concomitantly gave the highest accuracy and F1-score values of 93.20% and 92.20%, respectively, on the CCAT50 dataset. It recorded the highest F1-score of 76.69% and the second-best accuracy score of 63.40% on the Blogs50 dataset. In addition, it scored the highest accuracy of 86.48% with the second-best F1-score of 84.80% on the Twitter50 dataset. However, it came second in accuracy (96.50%) and third in F1-score (95.75%) on the IMDB62 dataset when compared to the 22 SOTA methods investigated. Its results are highly comparable with the top-performing methods with the highest accuracy score of 97.72 [1], and F1-score of 97.90 [87] on the IMDB62 dataset.

**Table 4.** Performance accuracy and F1 score of the RDNN and SOTA methods for PAA subtask.

| Methods | CCAT50 | | IMDB62 | | Blogs50 | | Twitter50 | |
|---|---|---|---|---|---|---|---|---|
| | Accuracy | F1-Score | Accuracy | F1-Score | Accuracy | F1-Score | Accuracy | F1-Score |
| Lexical and Topical [1] | | | **97.72** | 97.72 | | | 85.04 | |
| PMSVM [19] | | | | | | | | 70.00 |
| BertAA [35] | | | 93.00 | | 59.70 | 56.70 | | |
| SRNN [45] | 90.58 | | 94.10 | | 61.19 | | | |
| SCAP [45] | | | 94.80 | 94.80 | 41.60 | 41.60 | 82.50 | 82.50 |
| Imposters [50] | | | 49.90 | 76.90 | | 22.60 | | 52.50 |
| SVM Affix-punctuation 3-g [51] | 69.30 | | | | | | | |
| SVM with frequent 3-g [52] | 67.00 | 70.30 | 81.40 | | | | | |
| Continuous n-gram (2,3,4) [60] | 72.60 | | 94.80 | | | | | |
| SCAP n = 5 and L = 100 [68] | | | | | | | 31.83 | |
| IAF, RLP with n = 4, L = 100 [68] | | | | | | | 51.05 | |
| Character n-grams and Word n-grams [71] | | | | | | | 55.50 | |
| Character n-grams [71] | | | | | | | 60.00 | |
| n-gram and POS-tag [72] | | | | | | | 53.20 | |
| IAF, RLP with n = 4, L = 1000 [74] | | | | | | | 64.54 | 77.13 |
| CNN word [75] | | | 84.30 | 84.30 | 65.70 | 43.00 | 80.50 | 80.50 |
| CNN character [75] | | | 91.70 | 91.70 | 49.40 | 48.10 | 73.20 | **86.80** |
| n-gram CNN [76] | 76.50 | | 91.21 | | 53.09 | | | |
| Syntax-tree CNN [77] | 81.00 | | 96.16 | | 56.73 | | | |
| DeepStyle [85] | 80.45 | | 90.51 | | | | 64.80 | |
| BRET [87] | | 66.20 | **97.90** | | | 33.50 | | |
| LDAH-S with the topic [80] | | | 72.00 | 72.00 | 18.30 | 18.30 | | 38.30 |
| **RDNN Method** | **93.20** | **92.20** | 96.50 | 95.75 | **63.40** | **76.69** | **86.48** | 84.80 |

The effectiveness of the RDNN method is further presented graphically based on the accuracy and loss curves for visual validation purposes. Figures 2–5 show the accuracy and loss curves for the CCAT50, IMDB62, Blogs50, and Twitter50 datasets, respectively. It can be observed from the figures that the proposed method generally has better performance in accuracy with faster convergence speed in the training and testing processes on all the datasets. However, complex interactions between different stylistic features with nonlinearities can be observed in the Twitter50 datasets. Although performance improvement is achieved with the RDNN method, the effect of certain features such as hashtags is highly indicative. This blurs the boundaries of the proposed method in understanding the writing style of numerous authors. The Twitter50 dataset reveals a bias in the following ways. We did not preprocess the input post but converted it to lower case to erode the character sensitivity effect of a possible discrepancy between upper- and lower-case letters. The RDNN intrinsically leverages the consequential effects of uniform resource locators, hashtags, and mentions of names in the posts to achieve high accuracy and F1 score. The celebrities and power-users used different vocabularies in posts, as reflected in Figures 2 and 5. This was, for instance, by frequently tweeting similar messages to enhance their brands in comparison to regular users or bloggers who were interested in news, as shown in Figure 4. However, this behavior makes it hard for the proposed method to consistently learn the writing style of an author. It would be prudent for future work to apply byte-pair encoding to investigate the stylistic differences between regular authors and celebrities with their impact on PAA methods [102].

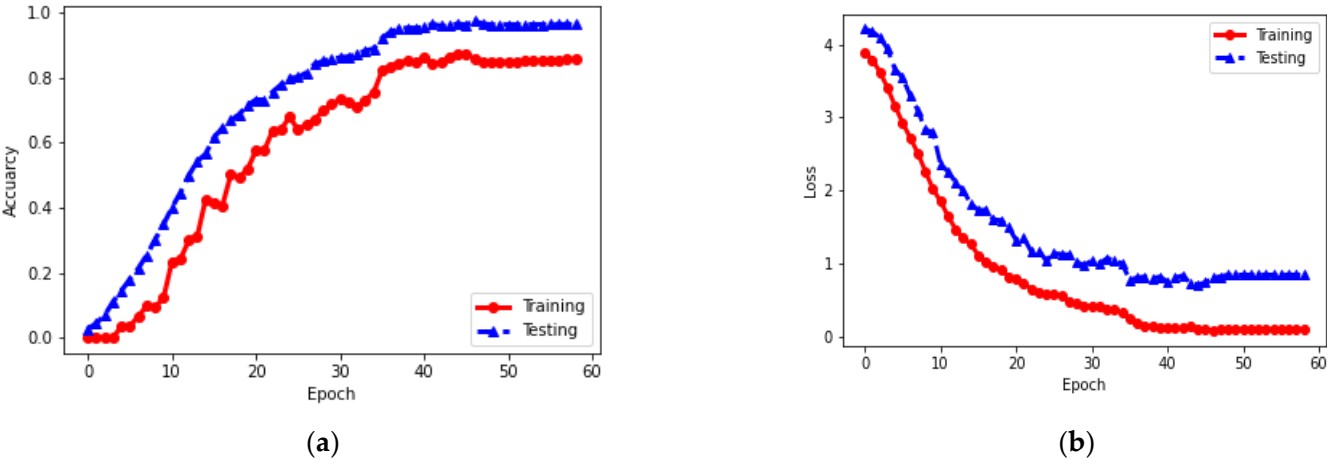

**Figure 2.** Performance accuracy and loss curves over epoch on the CCAT50 dataset: (**a**) accuracy; (**b**) loss.

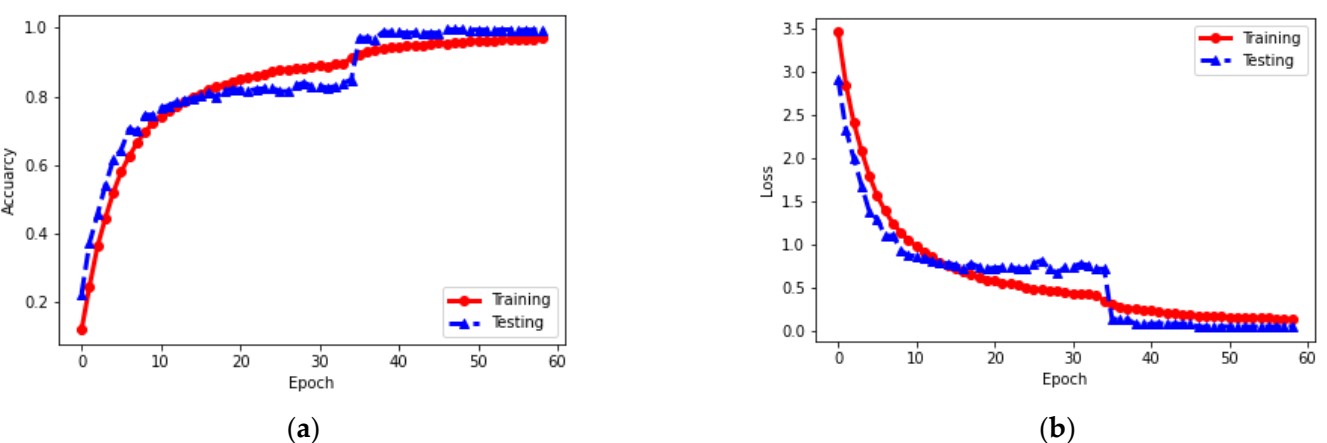

**Figure 3.** Performance accuracy and loss curves over epoch on the IMDB62 dataset: (**a**) accuracy, (**b**) loss.

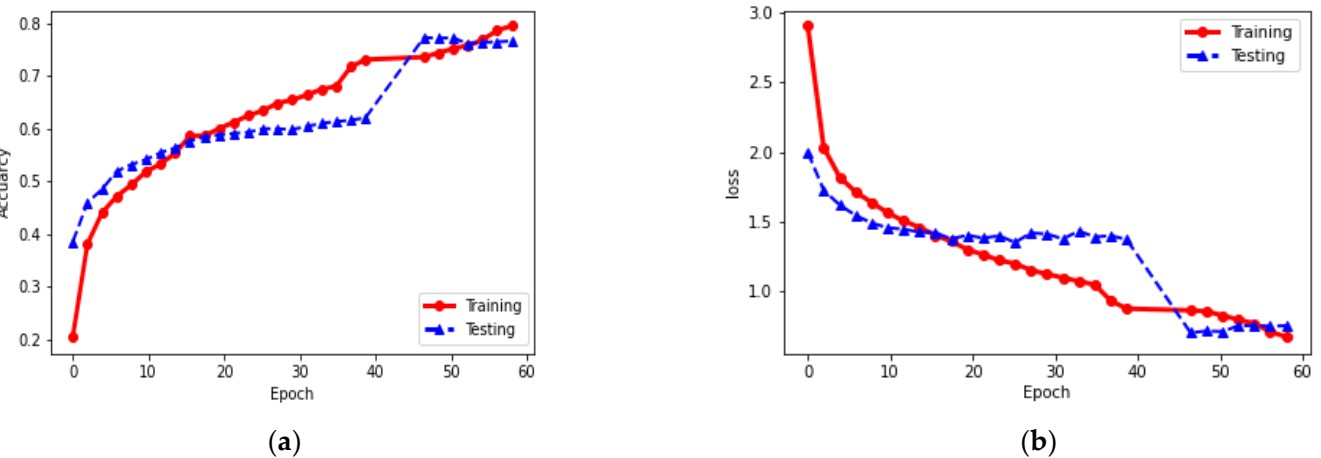

**Figure 4.** Performance accuracy and loss curves over epoch on the Blogs50 datasets: (**a**) accuracy, (**b**) loss.

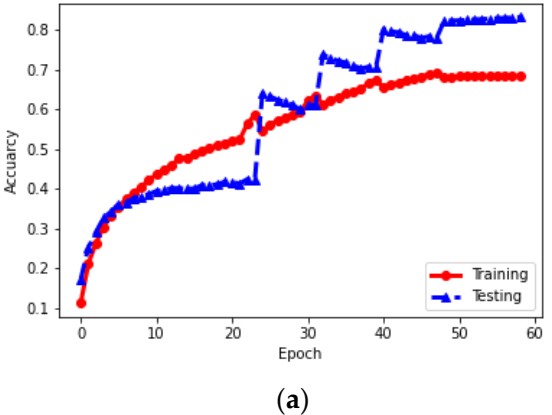

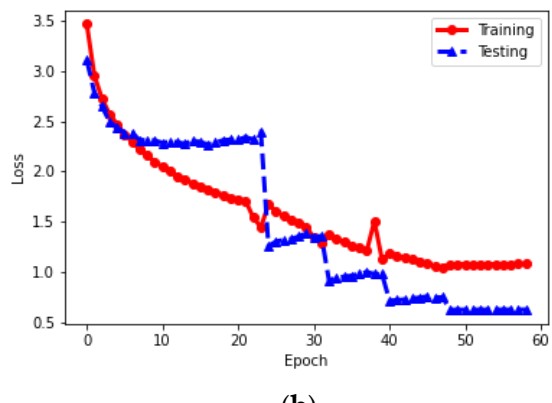

(**a**)  (**b**)

**Figure 5.** Performance accuracy and loss curves over epoch on the Twitter50 dataset: (**a**) accuracy, (**b**) loss.

The performance of the RDNN method was further tested to determine how author number and post length increased with fewer iterations for all the experimental datasets. Figure 6 demonstrates how the RDNN method performs in comparison to the SOTA methods on CCAT50 and IMDB62 datasets. The RDNN method has superior accuracy in three experimental datasets and achieves comparable performance in the fourth experimental dataset when compared to the other SOTA methods investigated.

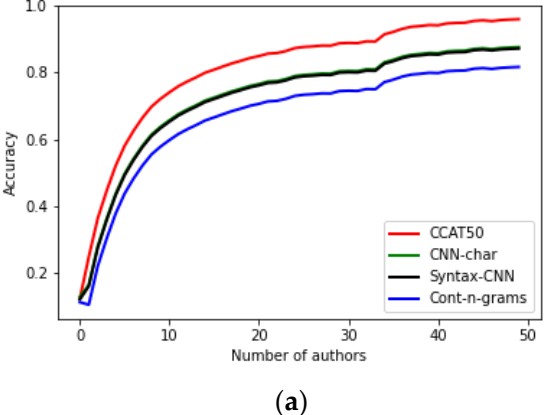

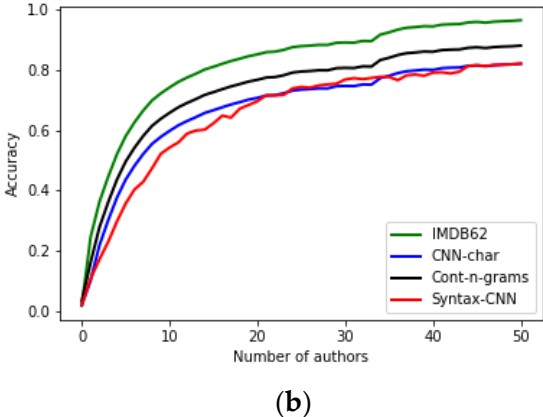

(**a**)  (**b**)

**Figure 6.** Performance accuracies on (**a**) CCAT50 dataset, and (**b**) IMDB62 dataset compared to SOTA methods with varying numbers of authors.

The results of several SOTA methods to analyze the stability of the RDNN method are reported for Blogs50 and Twitter50 datasets, as shown in Tables 5 and 6. Five independent runs were performed on Blogs50 and Twitter50 datasets. The maximum variability in accuracy is within 0.13%, to indicate that the RDNN method is still stable even if the precision is relatively low. The RDNN method can be postulated to be generally superior to the investigated SOTA methods on a large-scale dataset related to style-related tasks.

**Table 5.** Performance variance through repeated runs on the Blogs50 datasets.

| Metrics | Trial Number | | | | |
|---|---|---|---|---|---|
| | **1** | **2** | **3** | **4** | **5** |
| Accuracy (%) | 63.39 | 63.35 | 63.44 | 63.38 | 63.46 |
| Mean $\pm$ SD | 63.40 $\pm$ 0.045 | | | | |

SD: Standard deviation.

**Table 6.** Performance variance through repeated runs on the Twitter50 dataset.

| Metrics | Trial Number | | | | |
|---|---|---|---|---|---|
| | 1 | 2 | 3 | 4 | 5 |
| Accuracy (%) | 86.42 | 86.50 | 86.24 | 86.42 | 86.55 |
| Mean ± SD | 86.48 ± 0.051 | | | | |

SD: Standard deviation.

### 4.3.2. Result Visualization

The visualization of results is useful for demonstrating how the feature vectors obtained from the proposed method can support specialists in understanding the writing style of an author during the pretrial forensic investigation. The visualization analysis was performed using the LIME agnostic framework for interpretability [95]. LIME explains how the proposed PAA method made predictions by identifying the important input features. Figures 7–9 show the predictions from randomly selected sampled documents in the CCAT50, IMDB62, and Twitter50 datasets. In Figure 7, the proposed PAA method puts more weight on the author titled '*SamuelPerry*' and indicates a significant prediction probability on words such as *Hewlett*, *comput*, *price*, and *technology* with 41% contribution over the others. In Figure 8, the most important writing style to attribute the text sample to '*MartainWork*' is the web, internet, and company with 21%, while the other authors share approximately 59% of the example documents.

It was observed for the Twitter50 dataset in Figure 9 that the RDNN method assigned the same prediction probability of 32% to the authors "JacaNews" and "MaxBoot". In addition, the method put the same weight on the function words "american" and "britain", which represent two different classes of authors, not JacaNews and JacaNews. The correct decision of the method shows that features such as" america" and "wit" are from JaceNews as an author, and words such as "britain" are more significant to MaxBoot as the author with a probability value.

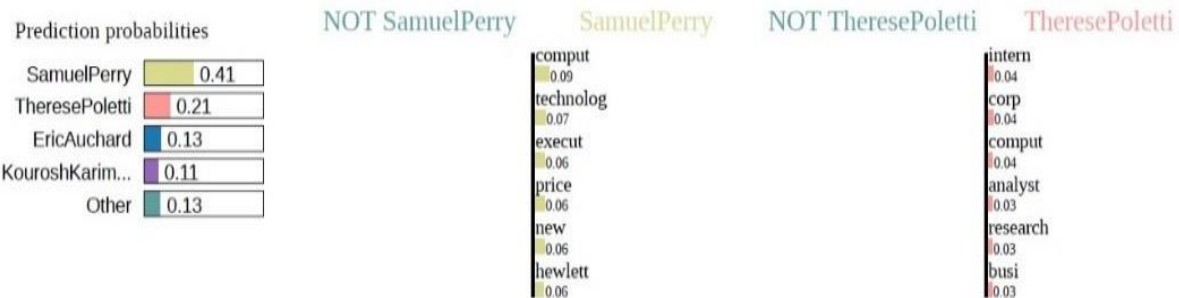

**Figure 7.** Explanation of individual prediction of the proposed method on the CCAT50 document using LIME.

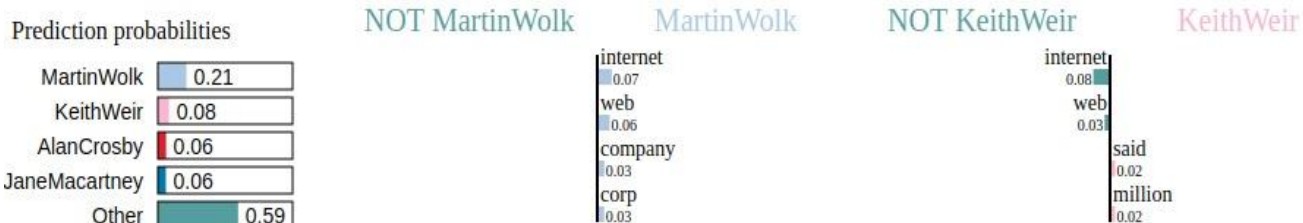

**Figure 8.** Explanation of individual prediction of the proposed method on the IMDB62 document using LIME.

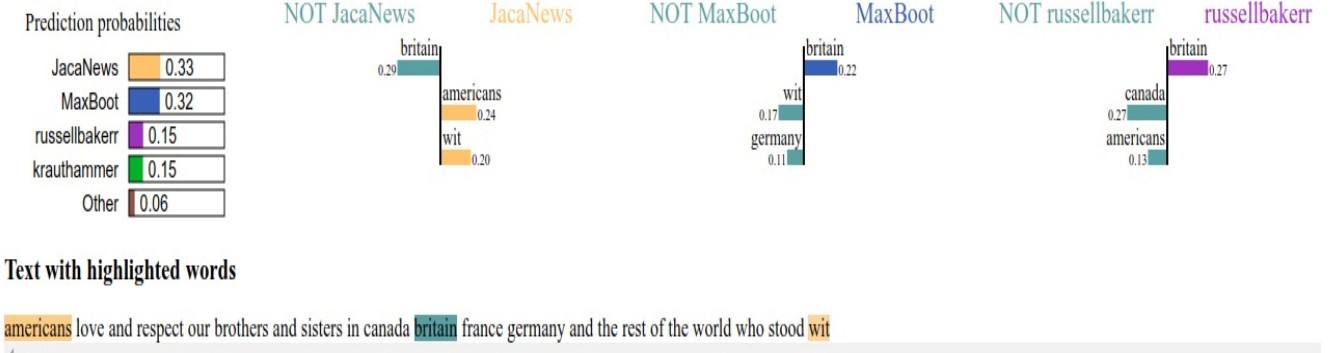

**Figure 9.** Explanation of individual prediction of the proposed method on the Twitter50 document using LIME.

The SHAP [94] over the feature vectors was utilized to obtain a good approximation of the SHAP values for the extracted features. The permutation was carried out by turning a particular word on and off. For example, if a word is not present in a particular post, turning it on could make a big difference to unveil the unknown writing style that could enhance the effective identification of an author. The nonexistent feature in the document can receive a high SHAP score. This was bypassed by applying permutations over the extracted features before the SoftMax. Figures 10 and 11 show the first random sample on the CCAT50 and IMDB62 datasets, and SHAP scores obtained by the proposed method can be attributed to the corresponding writing style selected over the training set. The features pushing the prediction higher are shown in red (predicting the correct author), and those pushing the prediction to lower (predicting the incorrect author) are shown in blue. However, this explanation could change as the input instance is altered. The SHAP local explanation considers only a specific instance at a time and generates an explanation by showing which feature values are making decisions toward the position and those that are negative. Figures 10 and 11 show local explanations where the probability of the output

is 86% for CCAT50 and 75% for IMDB62 datasets along with their writing styles or values are shown below ('*have*', '*own*', '*beginning*', .... '*getting*' .... '*some*') and so on.

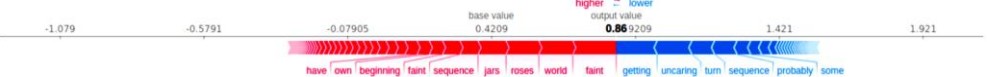

**Figure 10.** SHAP plot for local explanations of an instance on the CCAT50 dataset.

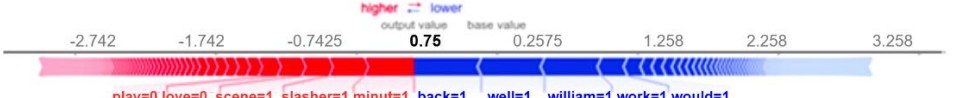

**Figure 11.** SHAP plot for local explanations of an instance on the IMBDB62 dataset.

Figure 12a,b show the SHAP summary plots for the global explanation of the model. The plots combine the features with important effects based on the sample texts from CCAT50 and IMDB62 datasets. Shapely value for a feature and particular sample is represented by a point on a summary plot. Features are on the *y*-axis and shapely values are on the *x*-axis. The colors are used to represent low or high values. Features are arranged according to their importance, and the peak feature in the summary plot is the most important, whereas the valley feature is the least important.

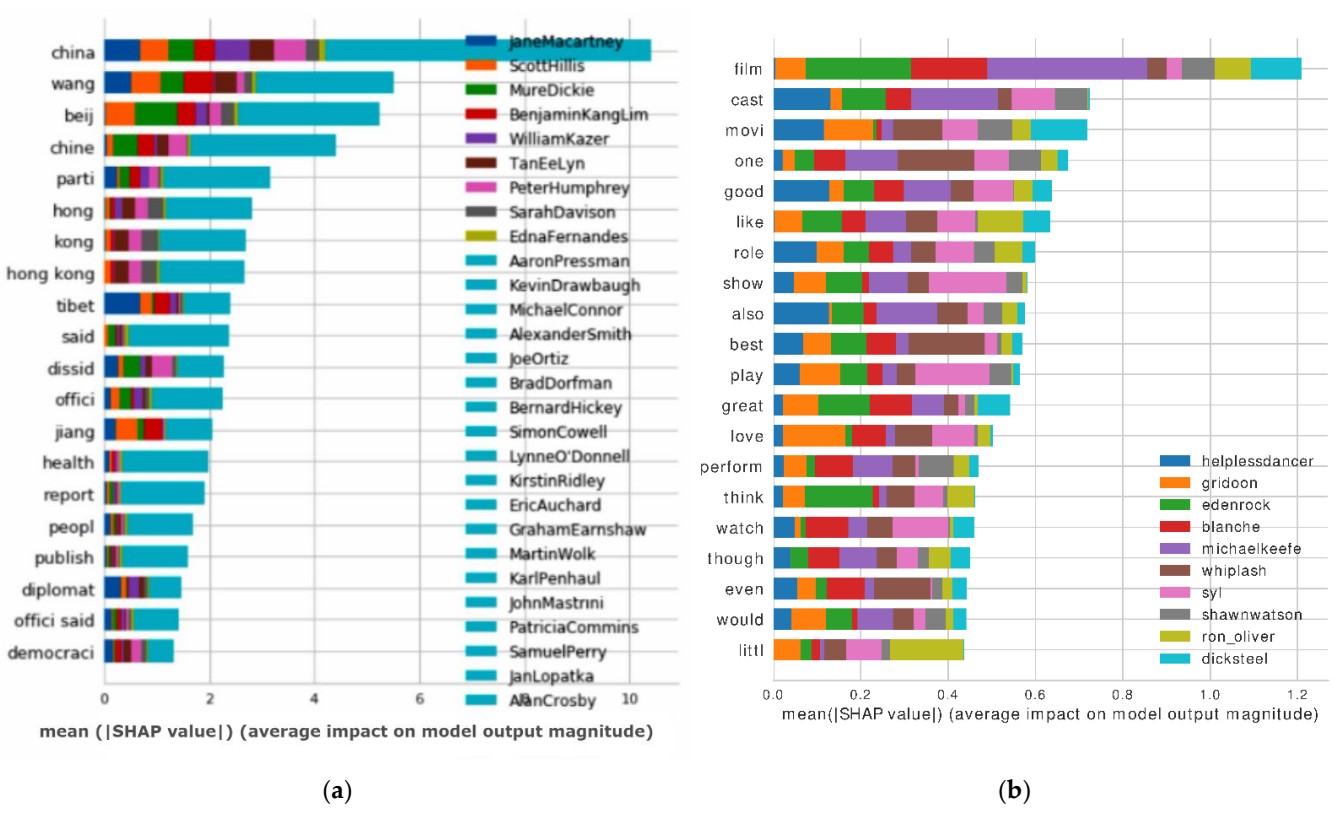

(**a**)  (**b**)

**Figure 12.** SHAP summary plots for (**a**) CCAT50 dataset, and (**b**) IMDB62 dataset.

## 5. Discussion

The best performance result for each SOTA method is here reported as presented in the corresponding articles. This is because the source codes of the comparative methods are unavailable. The proposed method outperforms most of the investigated SOTA methods, including the deep-learning methods, with an accuracy margin of 2.62%, 0.34%, and 3.98% on the CCAT50, IMDB62, and Twitter50 datasets, respectively. However, it is less superior to the top-performing method by 2.3% on the Blogs50 dataset. The accuracy of the method with the most frequent 3-g [52] did not perform well with the lexical, character,

and syntactic features on the CCAT50 dataset when compared to that reported in [51]. In contrast, the SCAP, LDAH-S, and CNN-word methods perform well in domains such as the CCAT50 and IMDB62 datasets where topical information is discriminatory. However, they achieved a comparatively worse accuracy for short messages such as the Blogs50 and Twitter50 datasets. It can be suspected that SCAP stores discrete n-grams for each author, which makes it simple to distinguish a topic preference for each author. However, it is unsatisfactory to capture the writing style of different authors or identify several posts written by an author. If such a correlation exists, the low accuracy reached by the CNN [77], SRNN [78], and DeepStyle [85] proves that the models carry less topical information. Moreover, the stylistic representation loses cohesiveness when the number of authors is very large.

The RDNN method is still superior in capturing better stylistic information such as words, topics, punctuation, or terms. In terms of the average F1-score, the RDNN method outperforms the comparative SOTA methods on CCAT50, Blogs50, and Twitter50 datasets by a margin of 21.9%, 19.9%, and 2.0%, respectively. However, the BertAA model is highly competitive on the IMDB62 dataset by roughly 2.15%. The word-based methods such as SVM, BertAA, CNN-char, and CNN-word perform well on CCAT50 and IMDB62 datasets, in which topical information is unprejudiced. However, our method achieves a comparative performance boost for Blogs50 and Twitter50 datasets. Even the shallow neural network of FastText embedding [60] does not beat the SVM histogram [46] with similar feature sets. Nevertheless, our PAA method shows superior encoding ability for the task of authorship attribution on both CCAT50 and IMDB62 datasets. The fine-tuned BERT, BertAA, and continuous n-gram representation model with pre-embedding vectors are sensitive to topical writing divergence between authors, which boosts performance on the IMDB62 dataset. However, they are less effective on the Twitter50 dataset than the CNN-char, where hashtags or emoticons are the most distinctive features. Moreover, our method outperforms the SVM model with frequent 3-g, LDAH-S, Imposters, CNN variants, and other deep neural-network-learning methods on three of the four datasets. This is in terms of accuracy and F1-score performance measures, except for the Twitter50 dataset, where CNN-char outperforms our PAA method in F1-score by 2% for 50 authors. The experimental results have demonstrated the aptitude of the RDNN method to handle the class-variation problem inherent in real-world text-communication applications [21].

The visualization plots of Figures 10 and 11 show the prediction for just one example of a testing dataset using the RDNN method. The SHAP values from the "based" value represent the features that are predicted from the RDNN method to reflect the writing style of an author. The analysis of how SHAP values are attributed to the writing style of an author is provided in Figure 12a,b. In this case, all features are continuous and vertically sorted by the average impact on the predictions. It can be observed that "*china*" and "*film*" have the maximum impact on the prediction with high mean values on CCAT50 and IMDB62 datasets. Similarly, high mean values recorded by "*wang*", "*beji*", "*cast*" and "*movi*" on CCAT50 and IMDB62 datasets are significant and related to the attributions of a writing style of an author in concordance with Figures 10 and 11.

## 6. Conclusions

Post-authorship attribution is still an open research problem because of different kinds of intrinsic ambiguities in text snippets. In this paper, a character-level authorship-attribution method is proposed based on a convolutional neural network, distributed highway network, and bidirectional long short-term memory. The proposed method can efficiently extract lexical, syntactic, and structural representations from a given post to identify the authorship of questionable texts. The proposed method has been extensively evaluated and compared against some of the state-of-the-art methods to report new results across the standard datasets. Some of the state-of-the-art methods performed comparably well in certain cases because they are designed with a simple structure, but they cannot be used as evidence in courts of law. At the same time, the proposed method eliminates

intrinsic fluctuations and makes high-quality interpretations. In contrast, we have shown how results obtained from the proposed deep-learning method can be interpreted using an explanation technique. Furthermore, a framework to identify the most influential stylistic features through a local explanation is provided that can be interpreted intuitively to give a useful tool to forensic investigators.

The present work presents certain inconspicuous impediments that should be considered in future studies to startlingly improve the performance of the proposed method. The effect of certain features such as hashtags blurs the boundaries of the proposed method in understanding the writing style of numerous authors. The possible user behavior of repeated tweeting of similar messages is hard for the proposed method to consistently learn the writing style of an author.

**Author Contributions:** Conceptualization, A.M. and T.C.; supervision, T.C. and V.M.; methodology, A.M. and O.O.O.; writing—original draft preparation, A.M.; funding acquisition, V.M.; writing—review and editing, O.O.O. All authors have read and agreed to the published version of the manuscript.

**Funding:** This research received no external funding.

**Institutional Review Board Statement:** Not applicable.

**Informed Consent Statement:** Not applicable.

**Data Availability Statement:** Not applicable.

**Acknowledgments:** The first author would like to thank the Department of Science and Technology (DST) and the Council for Scientific and Industrial Research (CSIR) for the inter-bursary support.

**Conflicts of Interest:** The authors declare no conflict of interest.

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
