# Peer review of "Post-Authorship Attribution Using Regularized Deep Neural Network"

_applsci, doi:10.3390/app12157518_

Round 1
Reviewer 1 Report
- The article needs an English review. For example, there are several long sentences in the Abstract.
- The motivation of the work is not string. Please make the motivation stronger in the Introduction section
- Related work section needs to be modified to summarize the previous studies with their results. For example, 10 references are listed in one line ( … and 101 structural [21-22,24,26,36,43-47]).
- What are the main limitations of related work and how are the authors going to address these limitations?
- Regarding the method in Figure 1, is this a new method proposed in this study or an old one?
- How are the hyperparameters of deep learning models tuned?
- The deep learning models need more details. How are they designed? What are the parameters used?
- It is good to study the complexity of the proposed models. There should be a tradeoff between complexity and accuracy
Author Response
Reviewer #1:
- The article needs an English review. For example, there are several long sentences in the Abstract.
Response
Thank you so much for your comments. The abstract section is thoroughly revised considering your valuable suggestion. Hopefully, you will find it logical.
- The motivation of the work is not string. Please make the motivation stronger in the Introduction section.
Response
Thank you so much for your suggestion. We have restructured the introduction with strong motivation in the revised manuscript for better understanding and to reflect the actual contribution of the study. Hopefully, now you will find it rational.
- Related work section needs to be modified to summarize their key features in previous studies with their results. For example, 10 references are listed in one line ( … and 101 structural [21-22,24,26,36,43-47]).
Response
We very much appreciate the suggestion, and we have updated related works with additional recent publications not only suggested by the reviewers but many others. The 10 references listed by the reviewers have been updated and are reflected in the revised version of the manuscript.
- What are the main limitations of related work and how are the authors going to address these limitations?
Response
We have taken this comment into full consideration, and the limitations of the previous work have been addressed thoroughly in the literature section, and how the limitations were addressed. We also provide a comparison based on common characteristics in Table 1. Hopefully, you will find it justified.
- Regarding the method in Figure 1, is this a new method proposed in this study or an old one?
Response
Figure 1 shows the proposed RDNN system crafted by us and well explained in this version of the manuscript.
- How are the hyperparameters of deep learning models tuned?
Response
Thank you so much for your valuable suggestion. The early stooping was used, and optimized hyperparameters for each dataset are described in Table 3. Because cross-validation is prohibitively expensive, we used a 5-fold cross-validation method with a Keras tuner for hyperparameter search as a pre-fold run time for the model with 10% randomly split off as a stratified test. This has been expressed thoroughly in section 4.2.
- The deep learning models need more details. How are they designed? What are the parameters used?
Response
Thank you so much for the suggestion. We have thoroughly made a clear description of the proposed RDNN method in Section 3 for improving the accuracy of the PAA system.
How are they designed?
Response
The study principally aimed to develop a regularized deep neural network (RDNN) for improving post-authorship attribution (PAA) accuracy. The method has been designed in a three-layer architectural PAA system (Figure 1). The system consists of a CNN character level layer, a distributed highway network (DHN) with bi-directional long short-term memory (BLSTM) DHN‑BLSTM layer, and a feature classification layer. The purpose of agglutinating the CNN character-level, DHN, and BLSTM was to achieve a PAA system with better classification performance, faster convergence speed, and high precision visualization capability in Section 3.
What are the parameters used?
Response
Thank you so much for the comment. We have entirely summarized the hyperparameter settings for each dataset in Table 3. We hope you will find it justified.
- It is good to study the complexity of the proposed models. There should be a tradeoff between complexity and accuracy?
Response
Thank you so much for the feedback. In the revised manuscript, we have added more information about the complexity of the proposed RDNN model from an accuracy and explainability perspective. However, the limited days given to us for revision do not favor the rerunning of codes to determine the running time of the proposed algorithm. We hope you will find it justified in Section 4.3.1.
Reviewer 2 Report
This paper presents an approach for post authorship attribution using a combination of different deep learning architectures (CNN, encoder-decoder BLSTM, highway nets and dense). Based on the results presented in the manuscript (Table 4 for instance) the proposed methods achieves better performance (accuracy and F1 score) in three out of 4 datasets explored.
The paper is well-written and the proposed method is interesting and valuable for research continuation. My recommendations to the authors are:
* the abstract is slightly confusing in terms of gain of the proposed method when compared with SOTA ones. Maybe it is better to be more specific, such as "the proposed approach obtained higher accuracy on 3 out of 4 datasets";
* the best F1 scores should be highlighted in Table 4 as well;
* in Figure 5 it can be seen that the loss of the test set is smaller than the training one at some epochs. What are the authors' explanation on why that happens?
* in the conclusions it is stated that although other methods achieved comparative good performance, they are complex. In which sense they are complex?
* in line 301, shouldn't it be "columns" of X^w instead of "column"?
* in line 79 it looks like there is a missing word.
Author Response
Reviewer #2:
- The abstract is slightly confusing in terms of gain of the proposed method when compared with SOTA ones. Maybe it is better to be more specific, such as "the proposed approach obtained higher accuracy on 3 out of 4 datasets?
Response
Thank you so much for the comment. The abstract section is thoroughly revised considering your valuable suggestion. Hopefully, you will find it justified.
- The best F1 scores should be highlighted in Table 4 as well.
Response
Thank you so much for the observation and valuable comment. We have highlighted the best F1 score in Table 4 of the revised manuscript.
- In Figure 5 it can be seen that the loss of the test set is smaller than the training one at some epochs. What are the authors' explanation on why that happens?
Response
Thank you so much for the observation and valuable comment. We have fully considered the comment and mentioned it in the revised manuscript in Section 4.3.1. Hopefully, you will find it justified.
- In the conclusions it is stated that although other methods achieved comparative good performance, they are complex. In which sense they are complex?
Response
Thank you so much for the comment. This sentence is restructured to make it more lucid for the reader.
- In line 301, shouldn't it be "columns" of X^w instead of "column"?
Response
Thank you so much for the comment. We have checked and the mistake is corrected in Section 3.1 as follows.
Here is the column ofand is the Frobenius inner product.
- In line 79 it looks like there is a missing word.
Response
Thank you very much for the suggestion, and sorry for the incomplete sentence. We have rephrased the wording. Hopefully, you will find it justified.
Reviewer 3 Report
This manuscript describes a regularised deep neural network (RDNN) developed by the authors with the purpose of identifying authorship of anonymously authored online text snippets such as news, blogs, and tweets.
The core of the manuscript, the RDNN and its context within the current landscape of authorship identification neural networks, is extensively and clearly characterized. In particular, Table 1 is an excellent summary of existing literature, and section 3 (“proposed method”) is concise and clear.
However, the introduction and discussion lack contextual depth.
Major points:
1) The introduction focusses largely on social media (especially twitter), while the datasets used for testing are primarily not what would be classically considered social media (news articles, movie reviews, blogs), with only one (Twitter 50) being directly relevant. This should be addressed by either:
- The addition of equal context justifying the utility of the RDNN in these other contexts in the introduction
- Highlighting the issue of generalizability of results to social media platforms in general in the discussion
2) The introduction almost unilaterally frames anonymity as a source of threat when misused, tacitly indicating that reverse-engineering text to identify authors is a positive endeavour. This needs to be balanced by consideration of ethical implications of subverting public fora with a social contract of anonymity, and for some social media sites the written contract guaranteeing privacy and anonymity to their users. Please either justify why this might not be appropriate for the current article, or add at least one paragraph exploring this point in the introduction, and highlight the need for further discussion around the ethics of using RDNN and similar algorithms in the discussion. Here are some citations to that may prove useful for thinking about this point – the authors are under no obligation to cite these particular works, they are simply a good starting point for thinking through this point.
Di Minin, E., Fink, C., Hausmann, A., Kremer, J., & Kulkarni, R. (2021). How to address data privacy concerns when using social media data in conservation science. Conservation Biology, 35(2), 437-446.
Bobicev, V., Sokolova, M., El Emam, K., Jafer, Y., Dewar, B., Jonker, E., & Matwin, S. (2013). Can anonymous posters on medical forums be reidentified?. Journal of medical Internet research, 15(10), e2514.
3) While it is understandable that the manuscript focusses on re-identification from text, the introduction would be strengthened by mentioning other competing approaches for similar forensic reidentification, such as exploring cross-site use of usernames in combination with time and date of activity (i.e. Zhou et al). This relates to the possibility that the RDNN approach may identify one author as the same person, but not provide any information as to who this person is.
Zhou, X., Liang, X., Zhang, H., & Ma, Y. (2015). Cross-platform identification of anonymous identical users in multiple social media networks. IEEE transactions on knowledge and data engineering, 28(2), 411-424.
4) Figure 4: Unlike the other figures, the training and testing line plots seem almost identically aligned. Yet, in table 4, the accuracy and F1 -score for the same test data (Blog50) is lower than the other test datasets. It is highly likely that the blue “testing” data presented is actually a duplicate of the “training” data. Please address.
Minor points:
Line 79: sentence is incomplete
Lines 89 through 94: the clear structure and use of headings makes this portion redundant. This could be removed.
Lines 273-275: The character and sentence limitation mentioned here is quite important, and should be raised in the discussion. Specifically, whether the RDNN model requires such limitations while other some of the most used other models do not.
Lines 286-289: this appears to be the inverse of the word stemming procedure common in natural language processing, where prefixes and suffixes are removed rather than added. If this is the case, the utility of the added complexity of adding this information is unclear, please explain. If this is not the case, please revisit this explanation so that others do not similarly misunderstand.
Line 307: the preceding explanation clearly relates to lexical and syntactic structure, but the link with topical structure is less clear. Is it the case that “topic” here refers to the most prevalent feature (e.g. a summary indicator of post content)? Please clarify, particularly given that the reader may confuse an output of the RDNN with topic analysis, which in the classical sense does not appear to be taking place.
Line 438: this is slightly unclear – by “model agnostic”, do the authors mean the visualisations can compare performance across models? If so, please use plainer language.
Line 444: word missing in “The proposed is seen to record good accuracy”
Line 469: “it is much better” is colloquial, please revise with more formal/precise language.
Line 475: Please specify “both standard deviations in percentage accuracy”
Pages 16 – 18: Possibly due to compression in the manuscript preparation, figures very low-resolution and accordingly very hard to read (especially noticeable in figure 7, 8, 9,10, 11, 12).
The discussion is missing an overview of the strengths and limitations of the current study (i.e. a strength is the comprehensive benchmarking of the new approach alongside alternative approaches; a weakness is the character and sentence limit).
The discussion is missing an integration with current and future practice, i.e. what further developments are required, or is the RDNN ready for real-world use even though performance might be improved? If so, in reference to issues raised in the introduction, what uses is it most suited for (or more importantly, MORE suitable for than currently available alternatives)?
Author Response
Reviewer #3:
Major points:
- The introduction focusses largely on social media (especially twitter), while the datasets used for testing are primarily not what would be classically considered social media (news articles, movie reviews, blogs), with only one (Twitter 50) being directly relevant. This should be addressed by either:
- The addition of equal context justifying the utility of the RDNN in these other contexts in the introduction
- Highlighting the issue of generalizability of results to social media platforms in general in the discussion
Response
Thank you so much for the comment and suggestion. The recommended citations provided insightful information in restructuring the abstract and introduction.
- The introduction almost unilaterally frames anonymity as a source of threat when misused, tacitly indicating that reverse-engineering text to identify authors is a positive endeavour. This needs to be balanced by consideration of ethical implications of subverting public fora with a social contract of anonymity, and for some social media sites the written contract guaranteeing privacy and anonymity to their users. Please either justify why this might not be appropriate for the current article, or add at least one paragraph exploring this point in the introduction, and highlight the need for further discussion around the ethics of using RDNN and similar algorithms in the discussion. Here are some citations to that may prove useful for thinking about this point – the authors are under no obligation to cite these particular works, they are simply a good starting point for thinking through this point.
Response
Thank you so much for the observation and valuable comment. We have mentioned it in the revised manuscript.
- While it is understandable that the manuscript focusses on re-identification from text, the introduction would be strengthened by mentioning other competing approaches for similar forensic reidentification, such as exploring cross-site use of usernames in combination with time and date of activity (i.e., Zhou et al). This relates to the possibility that the RDNN approach may identify one author as the same person, but not provide any information as to who this person is.
Response
Thank you so much for the observation and valuable comment. We have mentioned it in the revised manuscript.
- Figure 4: Unlike the other figures, the training and testing line plots seem almost identically aligned. Yet, in table 4, the accuracy and F1 -score for the same test data (Blog50) is lower than the other test datasets. It is highly likely that the blue “testing” data presented is actually a duplicate of the “training” data. Please address.
Response
Thank you so much for the observation and valuable comment. A correction is made in Figure 4 to capture the correct prediction of the proposed RDNN method for the Blog50 dataset.
Minor points:
- Line 79: sentence is incomplete
Response
Thank you very much for the suggestion, and sorry for the incomplete sentence. We have reworded the phrase for better understanding. Hopefully, you will find it justified.
- Lines 89 through 94: the clear structure and use of headings makes this portion redundant. This could be removed.
Response
Thank you so much for your suggestion. Yes, we have restructured the sentence for better understanding. Hopefully, you will find it logical.
- Lines 273-275: The character and sentence limitation mentioned here is quite important, and should be raised in the discussion. Specifically, whether the RDNN model requires such limitations while other some of the most used other models do not.
Response
Thank you for the suggestion. Yes, the weaknesses of the previous works have been thoroughly discussed, and show how the proposed method solves the problems in Section 2. We have discussed the limitations of our proposed RDNN model in the discussion and conclusion sections.
- Lines 286-289: this appears to be the inverse of the word stemming procedure common in natural language processing, where prefixes and suffixes are removed rather than added. If this is the case, the utility of the added complexity of adding this information is unclear, please explain. If this is not the case, please revisit this explanation so that others do not similarly misunderstand.
Response
Thank you so much for the suggestion. Yes, we have removed the prefix and suffixes for better understanding. Hopefully, you will find it logical.
- Line 307: the preceding explanation clearly relates to lexical and syntactic structure, but the link with topical structure is less clear. Is it the case that “topic” here refers to the most prevalent feature (e.g., a summary indicator of post content)? Please clarify, particularly given that the reader may confuse an output of the RDNN with topic analysis, which in the classical sense does not appear to be taking place.
Response
Thank you so much for the suggestion. Yes, we have restructured the sentence in the revised manuscript.
- Line 438: this is slightly unclear – by “model agnostic”, do the authors mean the visualisations can compare performance across models? If so, please use plainer language.
Response
Thank you so much for the observation and valuable comment. We have revised the manuscript.
“The visualization analysis was performed using SHAP [70] and LIME [71] for interpretability.”
- Line 444: word missing in “The proposed is seen to record good accuracy”.
Response
We noted the recommendation and incorporated it into the revised manuscript.
- Line 469: “it is much better” is colloquial, please revise with more formal/precise language.
Response
Thank you so much for the suggestion. Yes, we have restructured the sentence for better understanding. Hopefully, you will find it logical.
- Line 475: Please specify “both standard deviations in percentage accuracy”
Response
Thank you for the comment, we have included the standard deviation in Tables 5 and 6 as suggested.
- Pages 16 – 18: Possibly due to compression in the manuscript preparation, figures very low-resolution and accordingly very hard to read (especially noticeable in figure 7, 8, 9,10, 11, 12).
Response
Thank you very much for the valuable suggestion. We have replaced it with a new figure with better resolution. Hopefully, you will find it justified.
- The discussion is missing an overview of the strengths and limitations of the current study (i.e., a strength is the comprehensive benchmarking of the new approach alongside alternative approaches; a weakness is the character and sentence limit).
Response
Thank you very much for the valuable suggestion. This has been included in the revised manuscript. Hopefully, you will find it justified.
The discussion is missing an integration with current and future practice, i.e., what further developments are required, or is the RDNN ready for real-world use even though performance might be improved? If so, in reference to issues raised in the introduction, what uses is it most suited for (or more importantly, MORE suitable for than currently available alternatives)?
Response
Thank you very much for the valuable suggestion. This has been included in the discussion and conclusion sections of the revised manuscript. Hopefully, you will find it justified.
Round 2
Reviewer 1 Report
The authors have addressed my comments